

# TechMark: a framework for the development, engagement, and motivation of software teams in IT organizations based on gamification

Iqra Obaid and Muhammad Shoaib Farooq

Department of Computer Science, University of Management & Technology, Lahore, Lahore, Punjab, Pakistan

## ABSTRACT

In today's fast-moving world of information technology (IT), software professionals are crucial for a company's success. However, they frequently experience low motivation as a result of competitive pressures, unclear incentives, and communication gaps. This underscores the critical need to handle these internal marketing challenges such as employee motivation, development, and engagement in IT organizations. Internal marketing practices aiming at attracting, engaging, and inspiring employees to use excellent services have become increasingly important. Internal marketing is attracting, engaging, and motivating employees as internal customers to utilize their quality services. Gamification has emerged as a significant trend over recent years. Despite the expanding use of gamification in the workplace, there is still a lack of focus on internal marketing tactics that incorporate gamification approaches. Thus, addressing the challenges related to employee motivation, development, and engagement is crucial. Therefore, as a principal contribution, this research presents a comprehensive framework designed to implement gamified solutions for software teams of IT organizations. This framework has been tailored to effectively address the challenges posed by internal marketing by optimizing motivation, development, and engagement. Moreover, the framework is applied to design and implement a gamified work portal (GWP) through a systematic process, including the design of low-fidelity and high-fidelity prototypes. Additionally, the GWP is validated through a quasi-experiment involving IT professionals from different IT organizations to authenticate the effectiveness of framework. Finally, the outclass results obtained by the gamification-based GWP highlight the effectiveness of the proposed gamification approach in enhancing development, motivation, and engagement while fostering ongoing knowledge of the employees.

## INTRODUCTION

The fourth industrial revolution has intensified the competition between organizations (*Elidjen, Hidayat & Abdurachman, 2022*). On the other hand, the contribution of information technology (IT) organizations to economic growth has attracted considerable attention in recent years (*Odhiambo, 2022*). However, IT organizations' high turnover is

Corresponding author
Iqra Obaid,
f2019288001@umt.edu.pk

becoming their biggest challenge (*Pallathadka et al., 2022*). Employee turnover refers to the drop in the number of employees over a certain period or a progressive reduction that occurs for various reasons other than terminating employees (*Lazzari, Alvarez & Ruggieri, 2022*). This increased turnover can be disruptive for the organization as it directly affects its financial development and growth (*McCartney, Chi In & Pinto, 2022*). This high turnover includes job dissatisfaction, low motivation, future studies, better job opportunities, higher studies, less engagement, and other reasons (*Pallathadka et al., 2022*). These challenges make human resource management more complex and critical, as its primary element is improved service quality (*Sharma, Ahmad & Singh, 2022*; *Balouei Jamkhaneh et al., 2022*).

Previously, the services marketing sector gained prominence within the corporate world during the 1970s, highlighting the significance of service quality. Later this sector supported the emergence of a new idea known as internal marketing (*Qiu, Boukis & Storey, 2022*). Internal marketing prioritizes the satisfaction of human resources by considering an organization's employees as internal customers (*Huang, 2020*). Figure 1 depicts both the components of internal marketing and the associated challenges. Traditional internal marketing methods encompass training, workshops, management participation, collaboration, and other strategies to make employees more interested and motivated at work (*Qaisar & Muhamad, 2021*). However, in today's advanced era characterized by rapid technological evolution and the need for innovative thinking, these methods have fallen short of enhancing user growth, motivation, and engagement (*Rafiq & Ahmed, 2000*). This situation often leaves employees disinterested in their work (*Srivastava & Goyal, 2021*).

Creativity and innovation have become crucial factors for success in today's competitive organizations, such as IT and other innovative sectors. Employees are expected to think outside the box, contribute fresh ideas, and adapt to ever-evolving technologies (*Mirghaderi, Sheikh Aboumasoudi & Amindoust, 2023*). Numerous prior research endeavors have revealed that substantial competitive pressures, demanding corporate expectations, and challenging work environments have led to a decline in the motivation of system developers to remain within the company. This trend stems from a deficiency in both motivation and employee dedication to the organization. Furthermore, these problems persist due to traditional work environments that often lack modern approaches to employee engagement and development. Issues such as unclear incentives, where employees are uncertain about the rewards for their efforts, and a lack of direct feedback, which deprives them of valuable guidance on their performance, contribute to their low motivation and engagement. Moreover, the absence of clear communication about organizational goals leads to disconnection and a sense of aimlessness among IT professionals, hampering their development. Additionally, communication barriers and a lack of team cohesion inhibit collaborative efforts and hinder their professional growth. Addressing these challenges is crucial for fostering a more supportive and motivating work environment conducive to retaining talented system developers.

Traditional internal marketing practices, while well-intentioned, may not align well with these challenges. The one-size-fits-all nature of workshops and training sessions

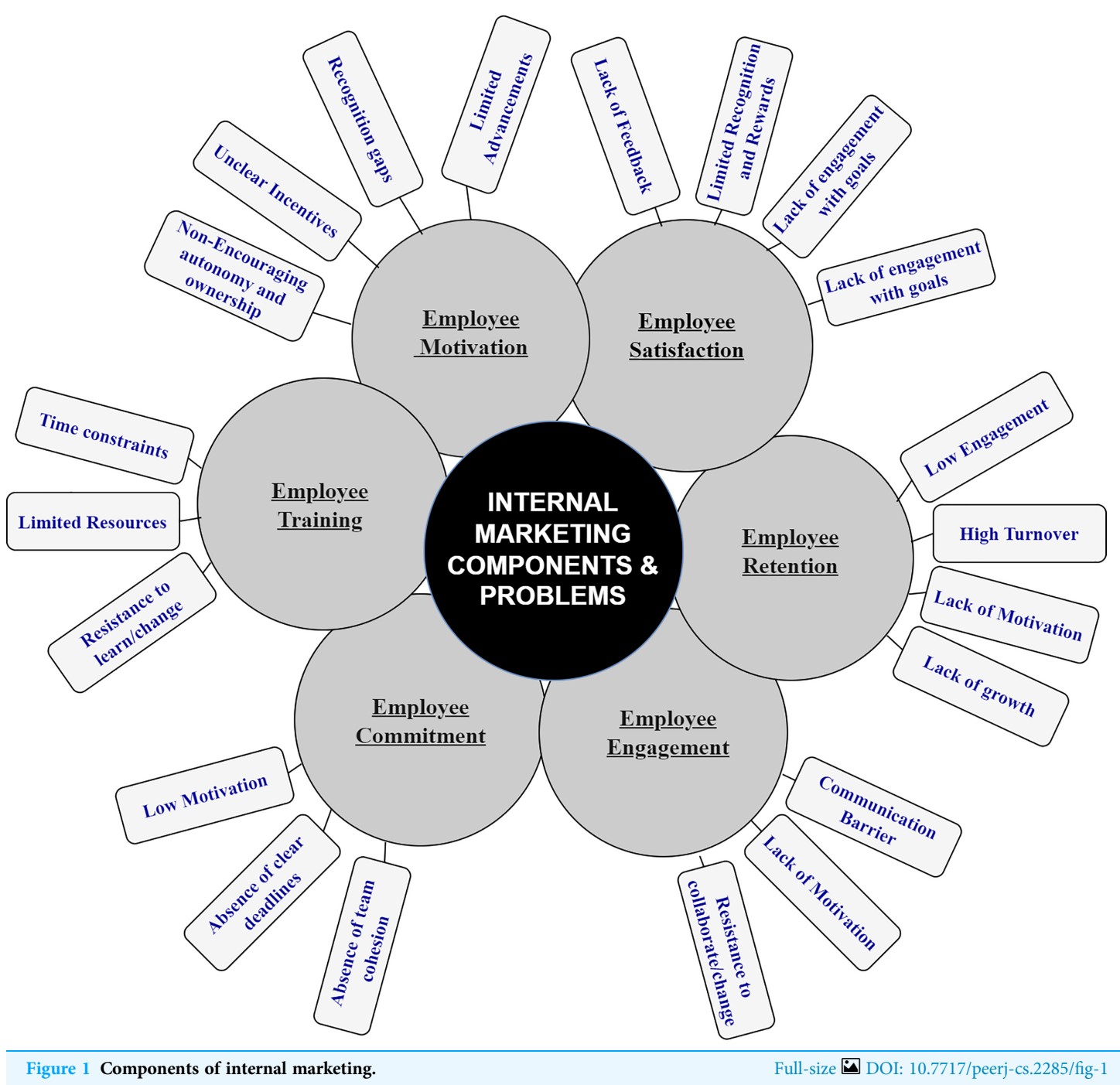

**Figure 1 Components of internal marketing.**

might not resonate with employees seeking personalized experiences. Additionally, the hierarchical management involvement might hinder employees' sense of autonomy and ownership in their work, consequently affecting employee motivation, engagement, and development (*Fraboni, Brendel & Pietrantoni, 2023*). This change compelled human resource management (HRM) professionals to embrace new methodologies and

employee-focused initiatives to motivate and engage their employees. In this context, gamification or serious games have recently gained attention among the HRM and the organization (*Georgiou, Gouras & Nikolaou, 2019*).

Gamification incorporates game features into non-gaming circumstances to increase user engagement, enjoyment, and motivation to accomplish a challenging and complicated task or reach a specific goal (*Deterding et al., 2011*). Considering these characteristics, gamification has attracted considerable attention in the workplace, both within and outside the organization. Moreover, gamification has gained popularity as a tool for increasing employee engagement and motivation in the workplace, thereby improving the organization's turnover (*Obaid, Farooq & Abid, 2020*). Several studies have demonstrated the effectiveness of gamification in enhancing employee performance, job satisfaction, and overall well-being because games play an essential role in our society, creating user motivation and involvement (*Bozkurt & Durak, 2018*; *Benitez, Ruiz & Popovic, 2022*).

The IT industry is a highly competitive and demanding sector that requires its employees to be constantly motivated and focused. Gamification stands out as a valuable strategy for boosting employee motivation and participation. The game mechanics tap into the human desire for achievement and recognition, thus promoting engagement and motivation (*Triantafyllou & Georgiadis, 2022*). Additionally, gamification can help employees develop new skills and knowledge, which is crucial in the rapidly evolving IT industry. In conclusion, gamification has become an increasingly popular tool for improving employee engagement, motivation, and attention in the work environment. With the rapid pace of technological change in the industry, gamification can help employees stay up-to-date with the latest trends and technologies while making work more enjoyable and rewarding.

The key objective of this research article is to introduce a comprehensive framework that implements gamified solutions for internal marketing practices within IT organizations. The framework is tailored to enhance motivation, development, and engagement within IT organizations, as leveraging gamification techniques can foster a more productive and engaging work environment. Moreover, the proposed framework facilitates the seamless integration of gamified techniques into diverse tasks within IT organizations. Furthermore, a dedicated task management portal is also created, validated, and accessed under the presented framework. Additionally, the portal underwent a rigorous evaluation through a quasi-experiment involving IT professionals from various IT organizations. The novelty of this work lies in its innovative framework and thorough evaluations. This study aims to provide valuable insights and answers to the following research questions in terms of the effectiveness of gamified applications in IT organizations:

1. What specific elements should be integrated into gamified frameworks to enhance motivation among software teams in the IT industry?
2. How can we conduct a robust evaluation to effectively evaluate the impact of gamified framework on the motivation and performance of software teams in their day-to-day tasks?

This research underscores the potential of gamification in boosting employee satisfaction, promoting teamwork, and driving innovation within IT organizations. The findings of this study offer valuable insights for IT organization managers and stakeholders interested in harnessing the power of gamification for organizational growth and improvement. This research study has been organized into several sections; "Related Work" provides extensive related work, examining prior studies on the influence of gamification on motivation and engagement; "Serious Game Design Model For IT Organizations" presents the proposed framework for implementing gamified solutions in IT organizations. Moreover, in "Serious Game Portal for Workforce Optimization", a gamified task management portal was designed and developed, and later in "Methodology", an experiment was set up to identify the validity of the presented framework. "Results and Discussion" presents the results and the discussions. Finally, "Conclusion" concludes the article.

## RELATED WORK

The concept of gamification has attracted substantial interest in recent times to enhance motivation and engagement in the work environment. This literature review aims to provide an overview of the existing research on the effectiveness of gamification. It will explore the benefits of gamification in terms of motivation, engagement, and employee performance.

*Dicheva et al. (2015)* systematically mapped gamification in education and found that gamification positively affects motivation, engagement, and learning outcomes. The study provides an overview of cutting-edge research in gamification and highlights the need for further research to explore the most effective gamification strategies in education. Nevertheless, the research did not put forth any concrete framework or model to substantiate the integration of game elements within the educational context. *Silic et al. (2020)* presented a study highlighting gamification's impact in the workplace. The study examined a gamified HRM system and its effect on employee engagement and satisfaction. The study considered 398 employees as a sample and showed that gamification positively affected employee attitudes and behavior. However, the study not only presents a gamified HRM system but it is also essential to note that the research focused solely on a specific organization as a case study, which has replaced its existing system with a gamified one. This narrow scope may introduce potential bias in the findings, as other gaming elements not explored in this study may yield diverse impacts on employee satisfaction and engagement. Hence, it is imperative to expand the research scope by conducting comparative studies across diverse organizations. Such an approach would facilitate a deeper understanding of the effectiveness of gamification in varied contexts, shedding light on its implications for employee motivation and performance. Moreover, broadening the scope would help mitigate potential data biases inherent in single-case studies, ensuring more robust and reliable findings.

*Gerdenitsch et al. (2020)* investigates the capacity of gamification as a tactic to amplify enjoyment and productivity within work settings. The research encompassed an online survey comprising 114 employees who utilized Habitica, a habit-tracking game, to gamify

their work-related responsibilities. The findings indicated that the implementation of work gamification positively impacted work enjoyment and productivity, particularly among employees in leadership roles. However, as the results are derived from a specific experimental procedure aimed at training managers in a bank, it is crucial to acknowledge that they may not be universally applicable to every task and work environment since different work environments possess unique dynamics that can influence the outcomes differently. Hence, there is a pressing need to investigate a wider range of games and applications to comprehensively grasp the impact of gaming elements in various work contexts.

*Friedrich et al. (2020)* explore gamification's capacity to boost employee motivation for knowledge sharing within knowledge management systems (KMS). The analysis shows that gamification can increase motivation for KM activities. Still, it requires a fitting environment, including an appropriate corporate culture and organizational climate that promote open knowledge exchange and rewards for KM activities. The article discusses motivations that support KS and KM activities, potential barriers to motivation, and gamification components to support KM activities. However, the study failed to implement any gamified model or framework to enhance employee motivation.

*Moldon, Strohmaier & Wachs (2021)* undertook a study to investigate how the behavior of software developers on GitHub changed after the removal of gamification elements. Eliminating streak counters caused behavior changes, cautioning online platform designers about gamification's unforeseen effects. However, the research neither presents any concrete framework to increase motivation nor evaluates the effect of gamified systems on all the employees of the IT organization.

*Hainey et al. (2011)* evaluated the effectiveness of a game in teaching software engineering requirements collection and analysis. They found that it positively impacted student motivation and learning outcomes. The study highlights the potential of gamification in enhancing the effectiveness of software engineering education. However, the study solely assesses the effect of gamified systems in teaching requirement gathering to students.

*Thelen et al. (2022)* investigate how gamification, specifically through leaderboards, boosts engagement with workplace-based assessment (WBA) tools in surgical residency programs. Weekly engagement intervention significantly increased WBA interaction, emphasizing the importance of engaging residents and faculty for targeted feedback and consistent evaluation in competency-based medical education. However, the study lacks a gamified framework to enhance motivation and engagement.

*Altomari, Altomari & Iazzolino (2022)* examines the growing importance of gamification in the labour market and its benefits, such as time-saving, data quality, and engagement. Moreover, AOC, a serious game, is designed to assess critical skills and provide feedback, highlighting positive results and user feedback. The study proposes the game's methodologies as an architecture for serious game design, emphasizing user journey and alignment with theory and assessment. However, the study fails to present any proper validation of the proposed architecture to prove its application and effectiveness in the practical environment.

Table 1 presents a detailed comparison of these studies, showing that gamification has shown promising effects in enhancing employee engagement, motivation, and productivity in different fields. However, careful design and implementation are necessary to avoid potential adverse effects. The increasing adoption of gamification suggests its potential as a tool for improving employees' performance in IT companies also. However, prior studies have overlooked IT organizations and the vital role of gamification in elevating intrinsic motivation and enhancing the motivation, development, and engagement of software teams. Recognizing this connection is crucial for maximizing team effectiveness within IT environments. A gamified framework can bridge these gaps by integrating game elements and incentives into the work environment, fostering a more engaging and collaborative atmosphere that enhances productivity and overall performance within the IT organizations.

However, the novelty of this research is to propose a framework for developing gamified systems tailored explicitly for IT organizations. The IT organizations, with their fast-paced and innovative work environments, present distinct challenges and opportunities compared to other industries. In such settings, where agility and creativity are crucial, traditional approaches may not suffice. Therefore, this framework aims to address the specific requirements and dynamics of software teams, ensuring that gamified systems are effectively implemented to enhance motivation, engagement, and performance. By focusing on IT organizations, this research seeks to fill a critical gap in the understanding of gamification's potential impact in the IT industry. Providing a structured framework aims to support IT organizations in effectively incorporating gamification elements into their operations, ultimately leading to improved performance and organizational success. Moreover, low and high-fidelity prototypes have also been designed against the presented framework. Furthermore, the framework is validated based on the designed prototypes.

## SERIOUS GAME DESIGN MODEL FOR IT ORGANIZATIONS

The primary research question is tackled through the creation of a framework tailored for IT organizations. This framework aims to facilitate the identification of specific elements crucial for boosting motivation among software teams within the organization. The implementation of this serious game framework helps the developers to create a unified and effective solution to the problem (*Jaccard et al., 2021*). Several studies have proposed serious game frameworks addressing educational, medical, and other problems (*Brauner & Ziefle, 2022*; *Marin-Vega et al., 2022*; *Rosenthal & Ratan, 2022*), but still, there is a serious need to draw attention to designing a gamified framework to address the issues of internal marketing in IT organizations. Therefore, this study proposed a serious game framework specifically tailored for IT organizations. The proposed serious game framework for IT organizations is founded on the principles of flow theory, aiming to create an optimal user experience that promotes engagement, concentration, and a sense of fulfillment. By incorporating flow theory, the framework seeks to enhance motivation, productivity, and collaboration within IT organizations (*Csikszentmihalyi, Montijo & Mouton, 2018*).

**Table 1  Comparative analysis of existing literature.**

| Ref. | Problem discussed | Model/Framework | Game or application designed | Limitation/Gap |
|---|---|---|---|---|
| *Dicheva et al. (2015)* | Education | x | x | The model was not defined. |
| *Silic et al. (2020)* | The lack of employee satisfaction and engagement in modern organizations. | Model defined: Integrating gamification into workplace processes can enhance employee job satisfaction and engagement. Model based on four major factors including enjoyment, recognition, gaming utility, and motivation is presented. | Application | The model and application-defined is not generic as it is limited to a single particular organization only |
| *Gerdenitsch et al. (2020)* | The effect of gamification on the enjoyment and productivity in the workplace | x | x | The model was not designed. |
| *Friedrich et al. (2020)* | The inadequate motivation of employees for knowledge sharing in knowledge management systems | x | x | No model was defined |
| *Moldon, Strohmaier & Wachs (2021)* | The effect of removal of gamification elements on software developers' behavior | x | x | The article did not define any model or implemented any application |
| *Hainey et al. (2011)* | Inadequate preparation of software engineering graduates for real-life scenarios as traditional teaching techniques are insufficient to help them learn | x | Game | The article only discussed the importance of game in learning requirement collection. No model was defined even in this regard |
| *Thelen et al. (2022)* | Difficulty in engaging surgical residents and faculty with workplace-based assessment (WBA) as traditional methodologies do not define its importance | x | x | The article did not define any model or implemented any application |
| *Altomari, Altomari & Iazzolino (2022)* | The lack of effective assessment tools for soft skills in the labor market | x | Application | The model was not defined |

## Flow theory

Flow theory, also referred to as the "flow state" or "optimal experience," originates from the work of Hungarian-American psychologist (*Csikszentmihalyi, 2000*). This theory aims to comprehend the state of optimal human experience and engagement, wherein individuals become fully absorbed in a challenging and rewarding activity. In this state, people often describe feeling completely absorbed, focused, and in control, losing track of time and self-consciousness. The key components and characteristics of flow theory are illustrated in Fig. 2. These components are integral to understanding the concept of flow:

Firstly individuals in the flow state exhibit intense focus which means they are intensely focused and paying close attention to their actions. This concentration level allows them to immerse fully in the task at hand, filtering out distractions and enhancing their

Obaid
Farooq
2024
10.7717/peerj-cs.2285

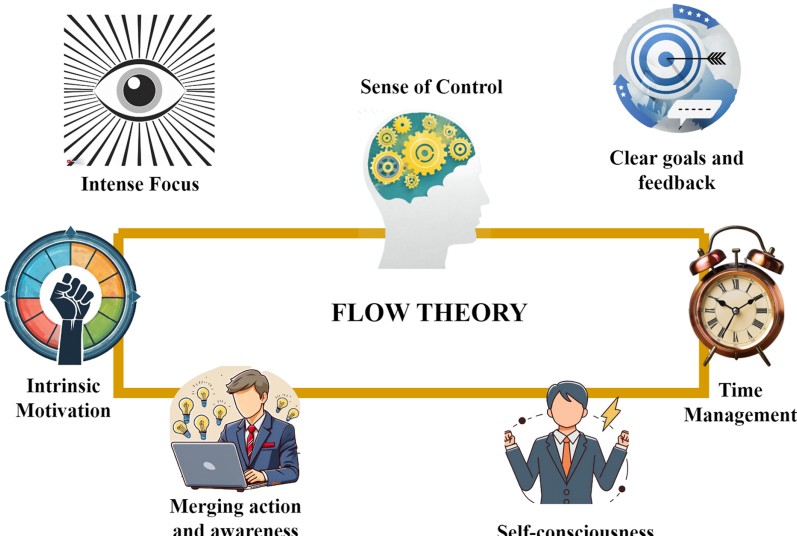

**Figure 2 Components of flow theory.** Images generated using AI, https://firefly.adobe.com/inspire/images.

performance. Secondly, a sense of control plays a pivotal role in achieving flow. When people feel in control of their actions and the outcome of the activity, they get flow experiences. Thirdly, clear goals and feedback are crucial aspects of the flow. The task has specific goals, and people get rapid feedback on how they are doing so that they may make necessary adjustments to their behavior. This ongoing feedback loop helps them stay engaged and make progress toward their objectives. Furthermore, flow experiences are often characterized by an altered sense of time. People in flow can lose track of time, and hours can fly by without their realizing it. This happens because they are so focused and absorbed in the activity. Moreover, flow is associated with a loss of self-consciousness. Individuals are free from self-doubt or concerns about external judgments. This absence of self-awareness allows them to focus entirely on the task and perform at their best. Additionally, flow involves the merging of action and awareness. In this state, the person becomes unified with the activity, and actions appear to unfold seamlessly without conscious effort. This unity of action and awareness makes the activity feel natural and smooth, as if it happens automatically, bringing great satisfaction. Importantly, flow activities are intrinsically motivating and fulfilling. Flow activities are intrinsically fulfilling and delightful, creating a sense of completion on an internal level. The intrinsic motivation derived from these activities encourages individuals to seek out and repeat the flow experience, fostering personal growth and well-being.

Csikszentmihalyi's studies on flow have had a broad impact in fields like sports psychology, education, and organizational psychology, helping individuals grasp how to attain their best performance and boost happiness in various aspects of life. Therefore, flow theory serves as the foundation for defining this gamified framework. By considering the principles of flow, such as intense focus, clear goals, and intrinsic motivation, in the design of a gamified system, IT organizations can create an engaging and rewarding experience for their employees. The decision to choose and incorporate flow theory into the design of

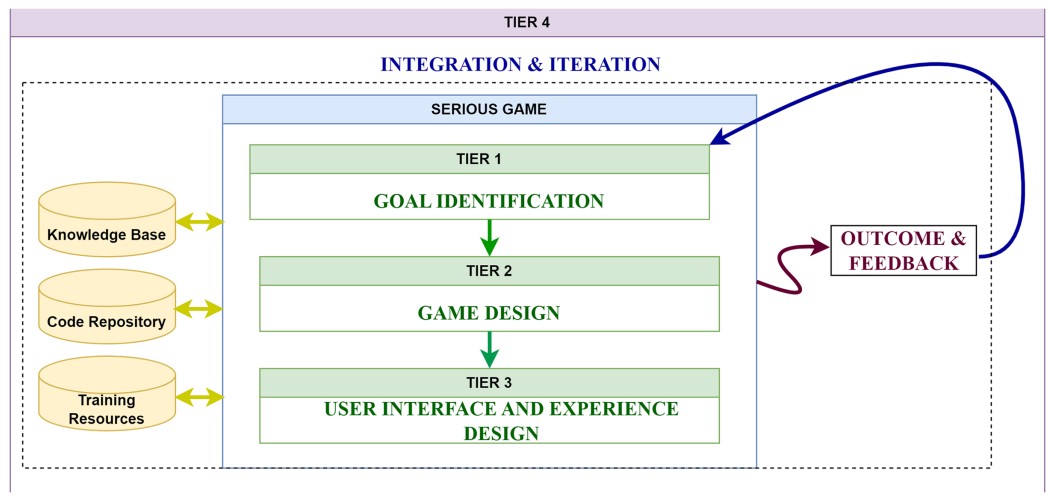

**Figure 3 TechMark: four-tier diagram.**

a gamified framework for IT organizations is rooted in the understanding that flow experiences can profoundly impact software professionals' motivation and performance within this unique work environment. In IT organizations, where individuals frequently encounter intricate tasks and rapidly evolving technologies, achieving a state of flow can be instrumental in enhancing job satisfaction, skill development, and overall engagement. By structuring the gamified framework to facilitate flow-inducing experiences, such as clear goals, immediate feedback, and a balance between challenge and skill, software professionals are empowered to immerse themselves fully in their work, leading to heightened creativity, sustained focus, and a greater sense of fulfillment in their roles within IT organizations.

## Proposed framework

The proposed framework TechMark consists of four tiers: goal identification, game design, user interface and experience design, and integration and iteration, as illustrated in Fig. 3. Each tier addresses specific activities and objectives, ensuring that all critical aspects of gamification are covered methodically. This structured division helps pinpoint challenges, design game by carefully applying gamification elements, develop user-friendly interfaces, and continuously refine the system.

Structuring the framework into distinct tiers, facilitates the effective management and integration of gamification elements. These tiers ensure that proper gamification elements are applied to address organizational challenges, which enhances employee engagement and productivity. Additionally, they ensure that the gamification strategy aligns with the organization's broader goals, leading to long-term success and sustainability. The elaboration on the components within each tier is provided below in the same section. Additionally, a thorough framework including all the components is presented in Fig. 4.

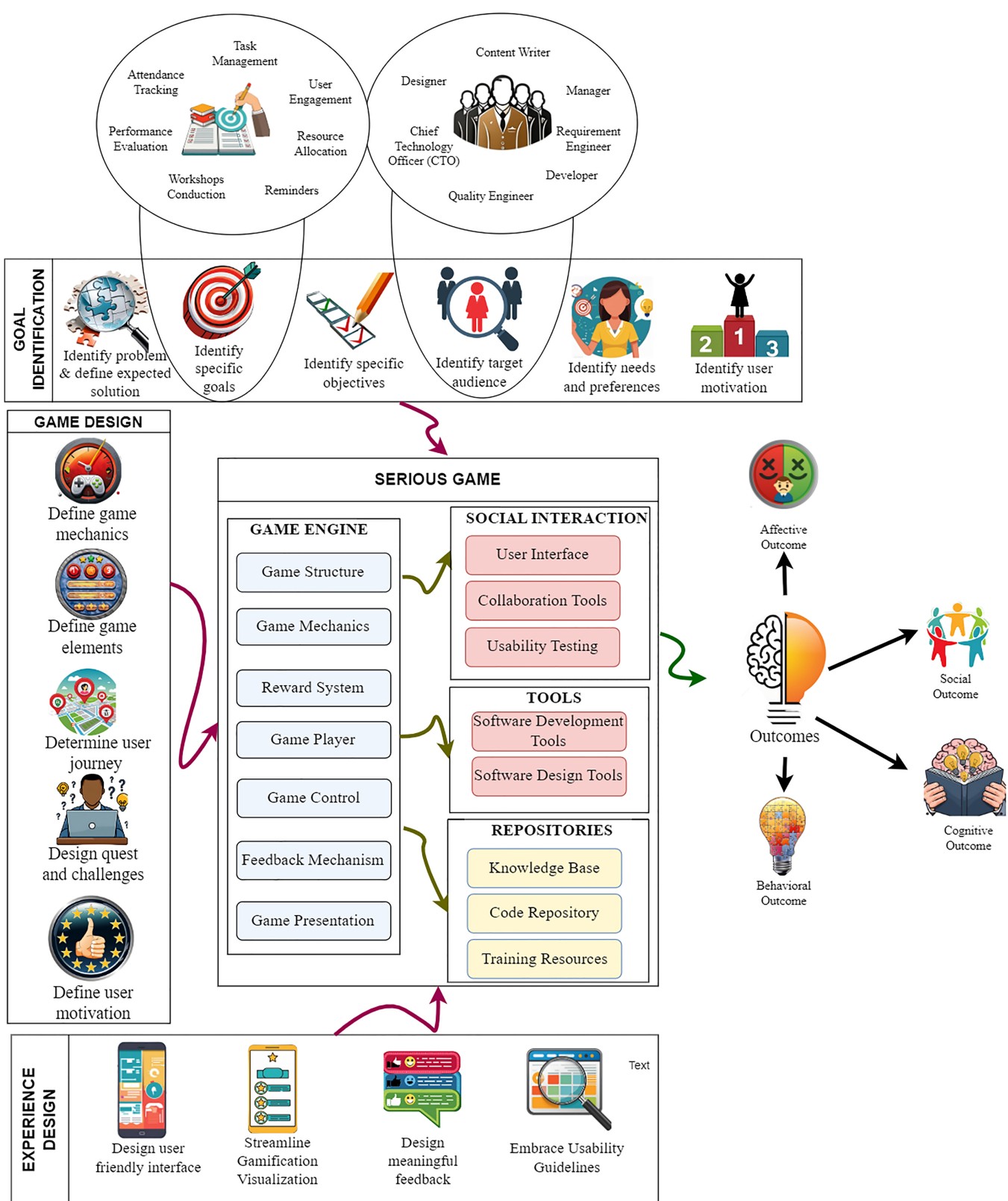

**Figure 4** TechMark: an internal marketing framework for IT organizations. Images generated using AI, https://firefly.adobe.com/inspire/images.

### Tier 1: goal identification

This foundational phase focuses on precisely defining the challenges within IT organizations that hinder motivation, development, and engagement. By identifying specific issues such as a lack of recognition, low team morale, or stagnant skill development, TechMark ensures that gamification elements directly address these problems, thereby enhancing employee engagement and motivation. Clear goals and objectives are established to align with the overarching objectives of the IT organization, ensuring that the gamified framework contributes directly to achieving organizational goals.

**Identify the problem and define the expected solution:** Correctly identifying the problem and defining an expected solution for the identified problem is one of the most important parts of the framework. Identifying the problem involves recognizing specific challenges or shortcomings within the IT organization environment that hinder team members' motivation, development, or engagement. These issues could include a lack of recognition for achievements, low team morale, inconsistent progress tracking, or stagnant skill development. This step ensures that gamification elements directly address existing problems, enhancing employee engagement and motivation. The framework optimizes collaboration, task handling, and overall software development processes by aligning the gamified approach with these specific issues. This targeted strategy results in measurable returns, elevated job satisfaction, and a well-designed gamified website that effectively addresses the IT organization's needs. Moreover, defining expected solutions entails designing strategies, mechanisms, or features within the gamification framework to address the identified problems. For instance, if the problem is a lack of recognition, the expected solution might involve implementing a badge and leaderboard system to showcase achievements. If stagnant skill development is the issue, challenges could be introduced to encourage continuous learning and improvement.

**Identify the specific goals:** Understanding the significance of identifying specific goals is crucial, especially within software teams of IT organizations, when customizing gamification strategies. These goals, including enhancing user involvement, optimizing project management efficiency, fostering developer skill development, and upholding high-quality software development, provide a roadmap for aligning gamification initiatives with the organization's overarching objectives. By articulating these goals clearly, IT organizations can tailor gamification approaches to address their distinct challenges and priorities, ensuring that the implementation of gamified systems effectively supports their strategic vision and organizational goals.

**Identify objectives:** Objectives are the actionable stages that help in achieving the larger organizational goals. In this phase, the TechMark framework deconstructs the identified goals into tangible, specific, and manageable parts. Each objective is clear, quantifiable, and has a defined timetable for completion, allowing for effective measurement of progress. For example, an objective may be to reduce the average project completion time by 20% over the following 6 months to enhance project management effectiveness. By breaking down the goals into measurable objectives, the TechMark framework enables IT organizations to

track progress and ensure that the gamification strategies contribute to achieving the desired outcomes.

**Determine the target audience and their needs and preferences:** Identifying the target audience within the IT organization is essential for customizing gamification features to resonate with their specific needs, preferences, and work dynamics. The TechMark framework recognizes that software professionals have unique requirements and motivations, and it aims to tailor gamification elements accordingly. By understanding the preferences and demands of the target audience, including developers, project managers, and other stakeholders, the TechMark framework ensures maximum user acceptance and efficacy in boosting productivity and engagement within the IT organization.

### Tier 2: game design

Tier 2, "Game Design," focuses on the foundational aspects of gamification within the TechMark framework. In this phase, the framework defines the critical game mechanics and elements, shapes the user journey, designs engaging challenges and quests, and understands and aligns user motivation. It sets the stage for creating a gamified IT environment that encourages participation, enhances productivity, and aligns with the preferences and motivations of the workforce.

**Define the game mechanics:** Identifying game mechanics is crucial while creating gamified software for IT organizations. The game mechanics define rules, incentives, and interactions, which influence user involvement and behavior. These game mechanics guarantees that the gamified website successfully encourages productivity, corresponds with the aims of the IT organization, and effectively motivates staff, thereby improving the user experience.

**Define game elements:** Defining game elements entails defining certain features contributing to the gamified experience, such as points, badges, levels, challenges, and leaderboards. These components combine to form a structured framework that promotes user interaction, develops healthy competition, and establishes precise goals for IT organization tasks. The gamified website may improve employee performance, teamwork, and motivation by introducing thoughtful gaming components, creating a more dynamic and effective work environment.

**Determine the user journey:** Defining a user journey is essential since it shows how users interact with the gamified product. The term "user journey" describes a user's sequential actions when utilizing a product or service, from the first interaction to accomplishing a particular activity or objective. The user journey in the context of a gamified website for an IT organization includes the full experience of developers and staff members as they go through various game mechanics, tasks, and challenges. Software companies may improve user pleasure, engagement, and the overall efficacy of their gamified platform by optimizing the user journey.

**Design engaging challenges, quests:** In a gamified system, quests are defined as particular tasks or missions, while challenges represent demanding objectives or hurdles for users to conquer. Within a gamification framework designed for IT organizations, these elements, quests, and challenges play a pivotal role by offering well-structured and

motivating employee targets. They not only elevate engagement but also stimulate skill enhancement, promote teamwork, and align individual endeavors with the IT organization's overarching goals. This boosts productivity and creates a dynamic and rewarding work environment, therefore enhancing overall performance and satisfaction.

**Define user motivation:** Understanding the intrinsic and extrinsic factors of motivations for people's participation and engagement in the gamified experience is necessary to define user motivation. It includes figuring out what motivates users whether a sense of achievement, rewards, competitiveness, knowledge, or social connection. The motivational elements of a gamified website for IT organizations must align with the employees' demands and preferences, which is why this phase is crucial. With multiple types of motivation, a gamification framework may successfully promote consistent participation and prolonged engagement, resulting in enhanced performance and happiness inside the IT organization environment.

### Tier 3: user interface and experience design

The "User Interface and Experience Design" phase focuses on creating a user-friendly interface and experience that maximizes engagement and satisfaction. For IT organizations, this entails streamlining gamification visualization, designing meaningful feedback systems, and embracing usability guidelines to ensure easy access and navigation for all team members. By prioritizing user comprehension, navigation, and overall satisfaction, TechMark promotes continuous participation and efficient task management within the IT organizations.

**Design user-friendly interface:** The term user interface (UI) describes the interactive and visual components that allow people to interact with a system, such as a dashboard of an IT organization's gamified website. It has a design layout, buttons, menus, and forms. Creating a user-friendly environment for an IT organization is essential in a gamified framework since it directly affects user engagement and experience. Tasks and interactions are more manageable with a well-designed UI because it improves user comprehension, navigation, and overall satisfaction. This step promotes continuous participation, efficient task management, and a positive perception of the IT organization's gamification initiatives by ensuring easy access to gamified features.

**Streamline gamification visualization:** A clear and thorough presentation of gamified features and progress monitoring is required for streamline gamification visualization. This calls for including user-friendly visual signals like progress bars, badges, leaderboards, and other graphical representations. The gamification components should be visually appealing and user-friendly for users to immediately understand their accomplishments, present situation, and the prizes they aim for. This stage is crucial in designing gamified applications because it keeps users interested in their progress, engaged, and motivated. This fosters a sense of achievement and promotes continuing engagement.

**Design a meaningful feedback system:** To design a useful feedback system, thinking of efficient ways to educate users about their actions and progress in a gamified environment is necessary. This can include timely alerts, visual signals, and messaging that let users know about their accomplishments, completed jobs, rewards gained, and overall progress.

Each user's successes should be considered while providing feedback, helping them feel valued and successful. Implementing such a system is crucial in a gamification framework for IT organizations as it reinforces positive behaviors, boosts motivation, and enhances user engagement by offering tangible evidence of their contributions and progress.

**Embrace usability guidelines:** The gamified website must be created with an emphasis on user-friendliness and accessibility to adhere to usability requirements. The goal of this stage is to design an interface that is intuitive, user-friendly, and accessible to a wide audience, including those with impairments. This is done by following accepted usability principles and usability standards. The gamified website is made more user-friendly and accessible to a wider variety of users by including elements like simple navigation, legible text, sufficient contrast, and support for assistive technology. This stage is important in a gamification framework for IT organizations since it ensures all team members can utilize the platform successfully, promoting a welcoming and engaging environment.

### Tier 4: integration and iteration

Tier 4, "Integration and Iteration," is the final phase in the TechMark framework, focusing on the practical implementation and continuous improvement of the gamified software. It ensures that the gamified solution aligns with user preferences, remains engaging, and effectively enhances the IT organization's workflow. This iterative approach drives ongoing improvement and user satisfaction within the gamified environment.

**Integrate the gamified elements into the software development workflow:** The seamless integration of gamified elements into the software development workflow includes integrating all formerly defined components, including social interaction features such as usability testing, collaboration tools, user interface design, and repositories encompassing knowledge bases, code repositories, and training resources. Moreover, integrate the commonly used software development and designing tools and platforms, such as version control systems (*e.g.*, Git), issue trackers (*e.g.*, Jira), and collaboration tools (*e.g.*, Slack). Ensure that the gamified features are accessible within the developers' existing workflows, minimizing disruptions and maximizing adoption. This comprehensive integration establishes a setting that improves user engagement, encourages efficient teamwork, assures user-friendly interfaces, and makes resources easily available. Combining these factors makes the software development process more engaging, effective, and collaborative, providing users with a more satisfying experience while promoting community and shared goals.

**Test the gamified software with a sample group:** After integrating gamified elements into the software development workflow, testing the gamified software with a selected group of users is essential. This sample group will interact with the system, providing valuable feedback on their experiences. This step helps identify any usability issues, technical glitches, or areas where improvements can be made. By gathering real user input, software developers can refine the gamification design to align more closely with user preferences and needs. Regular testing and feedback loops ensure that the gamified software evolves to maximize user satisfaction, engagement, and overall effectiveness.

**Iterate and refine the gamification design:** The gamification design must constantly improve, guided by user input and performance metrics. This iterative process ensures enhancements that align with users' evolving preferences and needs. Therefore, this process updates the gamified solution, adding new challenges, awards, and features contributing to its longevity and attractiveness. This dynamic strategy guarantees continuing participation and upholds the gamified solution effectiveness inside the workflow of the IT organization.

This four-tiered framework highlights the significance of gamification design aligning with IT organization goals, user experience comprehension, and ongoing gamification strategy refinement based on user input and data analysis. By adhering to this framework, IT organizations may create an effective and engaging gamified solution that motivates their staff and enhances the software development process.

## Validation of the model

The second research question is crucial for confirming the framework's effectiveness. It focuses on how to thoroughly evaluate the impact of the gamified framework on software team motivation and performance. The study employs the inter-rater reliability method (IRR) to validate the effectiveness and consistency of the proposed serious game framework (*Lange, 2011*). IRR measures the level of agreement between evaluators, with a score of 1 (or 100%) indicating complete agreement and 0% signifying complete disagreement. The study utilizes the inter-rater reliability approach to assess the effectiveness of the presented framework, providing full knowledge of the framework's impact and possible advantages. The validation results reinforce the framework's applicability and give significant insights for software companies looking to improve their development processes using serious game-based techniques. The gamified framework's validation process, employing the inter-rater reliability method, utilized the percentage acceptance agreement as the IRR measure. Through the consensus of several experts, it was determined that an agreement level of 75% to 90% was considered appropriate, indicating a substantial consensus among the evaluators in evaluating the proposed serious game framework (*Belotto, 2018*; *Md Tap, Mat Zin & Mohd Sarim, 2021*). This rigorous validation approach enhances the credibility and robustness of the gamified framework, ensuring its implementation in the IT organizations. Table 2 depicts the analyzed comments offered by the experts while validating the framework. The experts involved in this validation process include the researcher specializing in gamification and senior professionals from software teams within IT organizations. The experts emphasize the importance of incorporating the type of target audience, whether intrinsically or extrinsically motivated, into the framework. By considering the motivational orientation of the users, the gamified system can be tailored to better align with their specific needs and preferences. Therefore, the inclusion of audience segmentation is recommended as it ensures that the framework effectively caters to the diverse motivational factors of the users, optimizing their overall experience and engagement with the gamified approach.

**Table 2 Inter-rater reliability expert analysis.**

| No. | Components | Evaluators | | | AB | AC | BC | Level of agreement | No. of agreement |
|-----|-----------|---|---|---|----|----|----|--------------------|------------------|
| | | A | B | C | | | | | |
| 1 | Goal identification | 1 | 1 | 1 | 1 | 1 | 1 | 3/3 | 3 |
| 2 | Game structure | 1 | 1 | 1 | 1 | 1 | 1 | 3/3 | 3 |
| 3 | Game mechanics | 1 | 1 | 1 | 1 | 1 | 1 | 3/3 | 3 |
| 4 | Game player | 1 | 1 | 1 | 1 | 1 | 1 | 3/3 | 3 |
| 5 | Game control | 1 | 0 | 1 | 0 | 1 | 0 | 1/3 | 1 |
| 6 | Game presentation | 1 | 1 | 1 | 1 | 1 | 1 | 3/3 | 3 |
| 7 | User journey | 1 | 1 | 1 | 1 | 1 | 1 | 3/3 | 3 |
| 8 | Reward system | 1 | 1 | 1 | 1 | 1 | 1 | 3/3 | 3 |
| 9 | Feedback mechanism | 1 | 1 | 0 | 1 | 0 | 0 | 1/3 | 1 |
| 10 | Quest and challenges | 1 | 1 | 1 | 1 | 1 | 1 | 3/3 | 3 |
| 11 | Game elements | 1 | 1 | 1 | 1 | 1 | 1 | 3/3 | 3 |
| 12 | Experience design | 0 | 1 | 1 | 0 | 0 | 1 | 1/3 | 2 |
| 13 | Outcomes | 1 | 1 | 1 | 1 | 1 | 1 | 3/3 | 3 |
| 14 | Usability principles | 1 | 1 | 1 | 1 | 1 | 1 | 3/3 | 3 |
| 15 | Collaboration tools | 1 | 1 | 1 | 1 | 1 | 1 | 3/3 | 3 |
| 16 | Usability testing | 1 | 1 | 1 | 1 | 1 | 1 | 3/3 | 3 |
| | Total | | | | | | | | 42/48 |
| | Percentage | | | | | | | | 0.875 |
| | Inter-rater reliability mean: | | | | | | | | 87.50% |

# SERIOUS GAME PORTAL FOR WORKFORCE OPTIMIZATION

This section presents a gamified work portal (GWP) for workforce optimization in the IT organizations. The portal aims to monitor and improve the day-to-day target tasks and activities of employees in an IT organization to maximize their productivity and performance. By gamifying the process, employees are motivated to actively engage in their tasks and work towards achieving their targets, leading to better overall performance and efficiency within the organization. The portal strictly follows the components of the presented framework TechMark. The design process involves identifying the problem and gathering requirements from stakeholders, which is accomplished through a preliminary survey and comprehensive literature review. The next step is to design the interfaces and modules, considering the theoretical framework's elements, which ensures that the final design aligns with the identified problem and meets the specific requirements of the stakeholders. Later the game based-portal is developed and coded on a web platform. Furthermore, the evaluation of the portal was conducted in two stages.

## Low-fidelity prototype of GWP

This section presents the development of low-fidelity prototypes for the Serious Game Portal designed to optimize workforce performance in the IT organizations. The

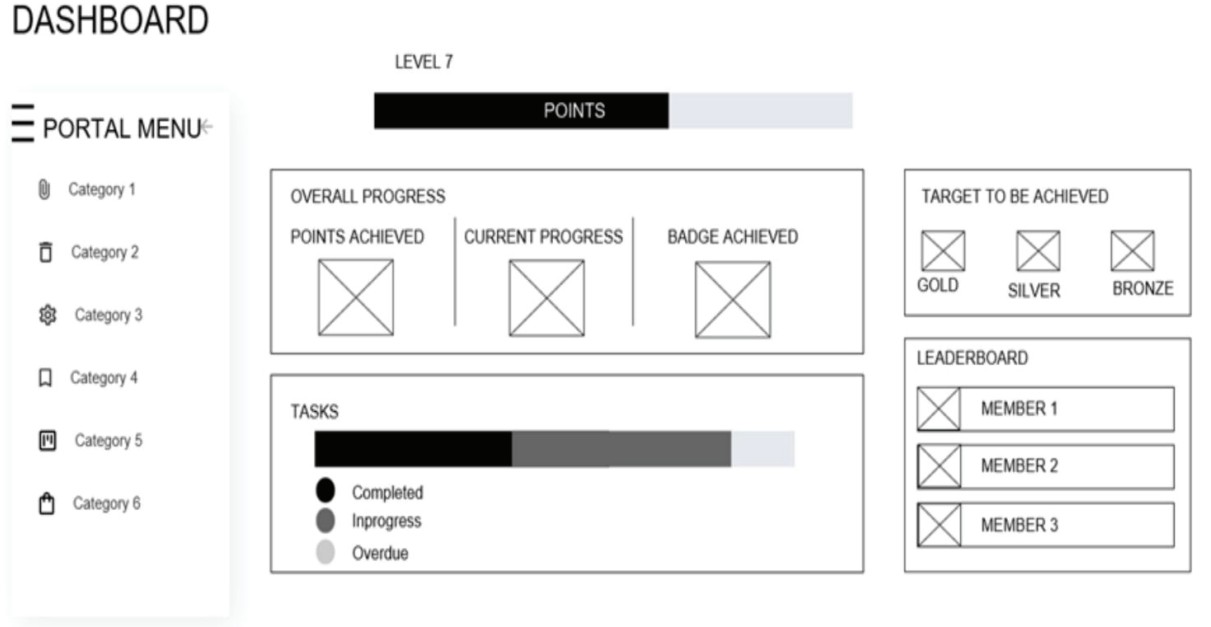

A

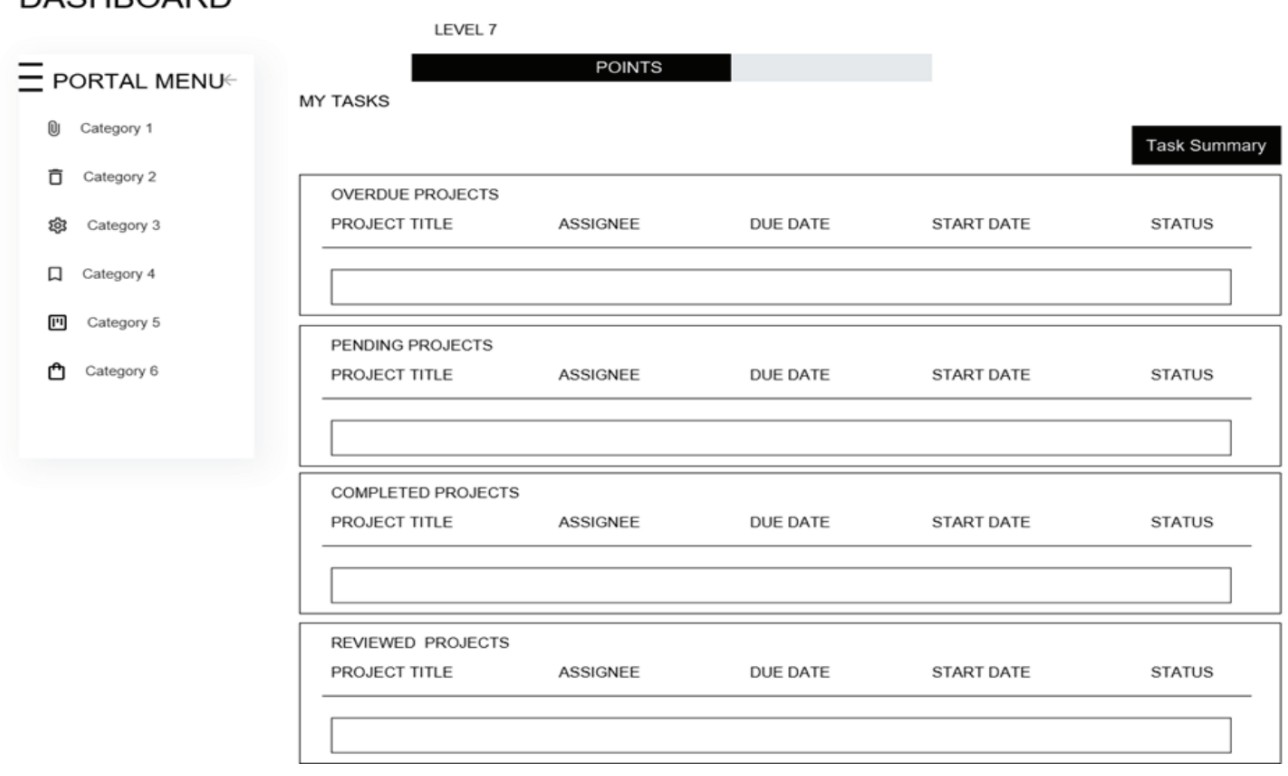

B

**Figure 5 Low-fidelity prototype of GWP-(A) Low-fidelity prototype of dashboard-(B) Low-fidelity prototype of task.**

theoretical framework presented above served as the foundation for the game's structure and objectives. The portal aims to enhance productivity and engagement among IT professionals through gamified task management. For instance, within the portal, team leaders assign tasks to developers, each with designated point values, in a unified scenario. As developers complete tasks and receive evaluations from team leads, they accumulate points, thereby achieving various badges upon reaching specific milestones. Figure 5 represents the low-fidelity prototype of the portal, where Fig. 5A is the dashboard prototype and Fig. 5B is the prototype of the task status screen.

## High-fidelity prototype of GWP

This section presents the high-fidelity prototypes of the IT organization optimization portal. The primary objective in creating these high-fidelity prototypes is to incorporate interactivity and user-friendly attributes that boost productivity and engagement among IT professionals. Upon accessing the portal, users are prompted to log in to their personalized accounts. The main page of their portal features a comprehensive dashboard showcasing their work progress, earned points, and achieved badges. Additionally, a leaderboard is incorporated to boost motivation, displaying top performers among the users. Moreover, the portal enables easy task tracking, empowering users to manage their assignments efficiently. The user-friendly interface ensures seamless navigation and provides valuable insights into their performance, ultimately fostering a more productive and engaging work environment. Figure 6 represents the high-fidelity prototypes of the GWP.

## METHODOLOGY

### Study group

This research study aims to assess the significance of the gamification framework in enhancing motivation, engagement, and development among teams and individuals in the IT organizations. A practical approach was adopted to achieve this objective by developing and deploying a framework-based gamified website as an example across five IT organizations specializing in providing web-based, mobile, and artificial intelligence (AI) solutions. The size of each company varies but ranges between 15 to 30 professionals, ensuring a diverse pool of expertise and resources for the project's successful evaluation. The purpose of creating the gamified website was to experiment with the effectiveness of gamification techniques within real-world software development environments. To achieve the objective, a Quasi-experimental research design (*Achen, 2021*) has been chosen. The research involved 50 software professionals, encompassing 20 developers, 15 designers, and 15 testers, with a division of 25 males and 25 females among them, all having at least 2 years of experience. Each participant furnished formal written consent, affirming the confidentiality of their personal information and acknowledging its intended research use, under Department of Computer Science, University of Management and Technology institutional review board guidelines, as per approval letter No. UMT-Reg/2022/32-1(1). The intervention group, comprising 25 software professionals, will be exposed to a gamified working environment to complete their day-to-day tasks. In

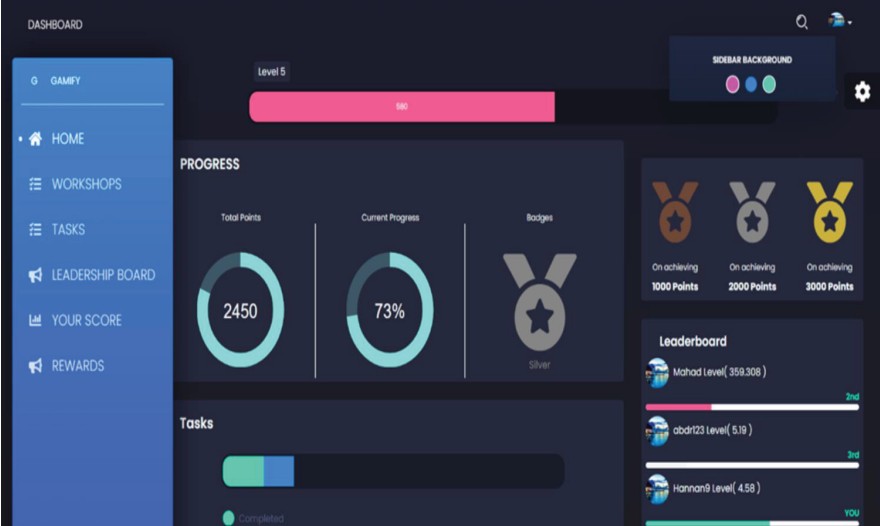

A

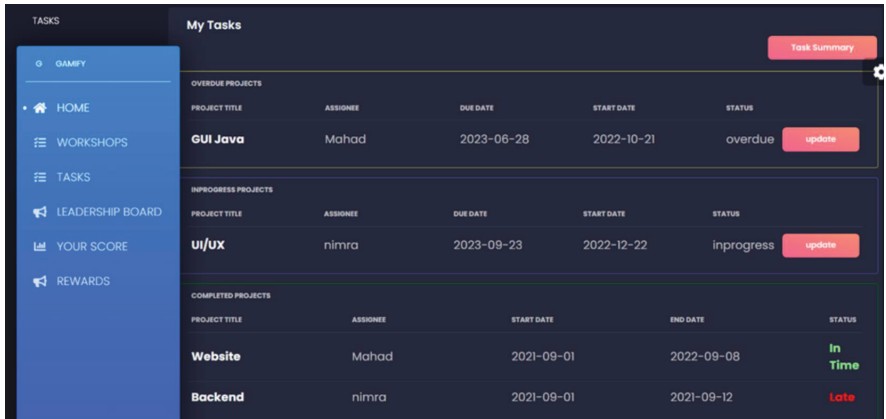

B

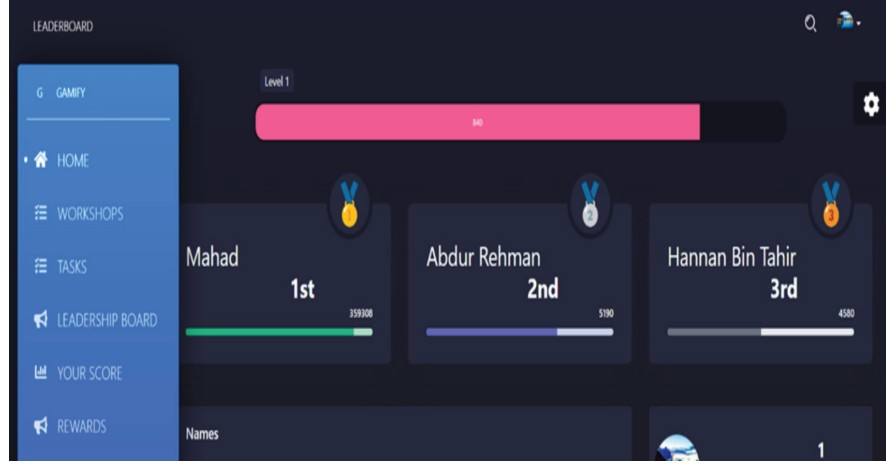

C

**Figure 6  High-fidelity diagram of SGP-(A) Dashboard-(B) Task details-(C) Leaderboard.**

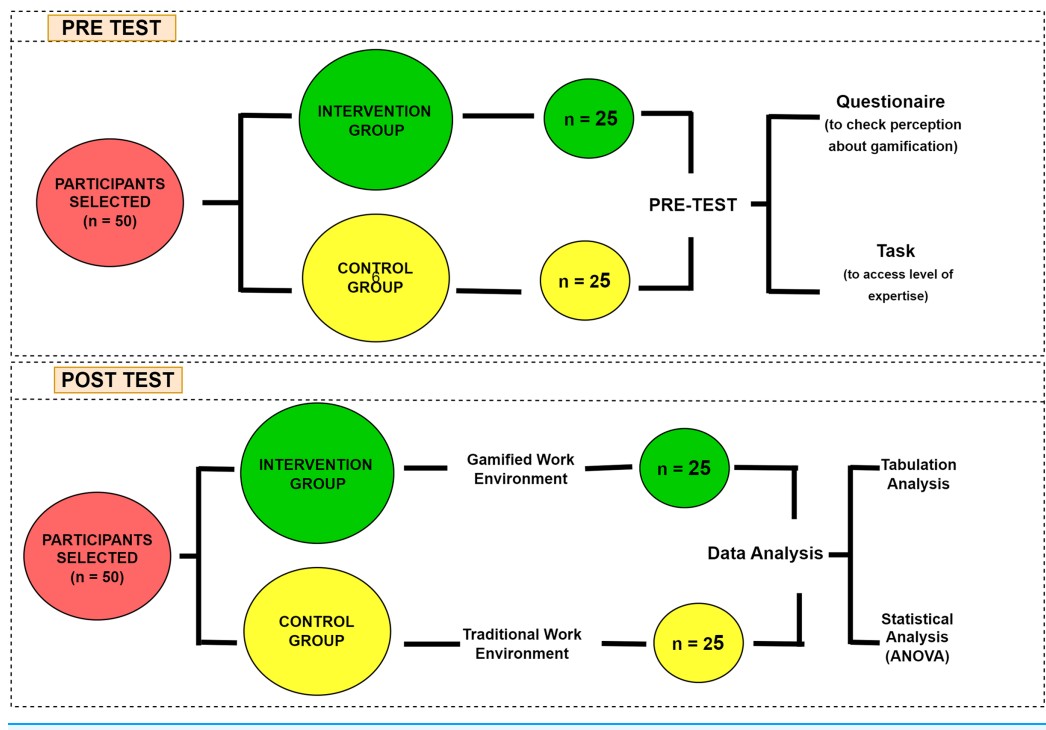

**Figure 7 Experimental procedure.**   

contrast, the control group of 25 software professionals will work in the traditional work environment. The experiment comprises two distinct phases: the pre-test and the post-test. Figure 7 depicts the comprehensive experimental procedure conducted in this research.

## Pre-test

In order to assess the software professionals' inclination and openness towards incorporating gamification into their daily work routines, as well as to ensure uniform expertise levels among all participants, a pre-test was conducted. The first phase of the pre-test encompassed a 10-question questionnaire designed to gauge participants' perceptions towards gamification, accompanied by an additional task aimed at assessing their level of expertise in the second phase. Table 3 presents the details of the results gathered through the questionnaire. The data from Table 3 underscores software professionals' favorable attitudes toward gamification (Mean = 4.02). They express a desire for workplace gamification (Mean = 3.90) and a readiness to adopt it (Mean = 3.98). They believe gamification enhances enjoyment, interaction, and participation, highlighting their readiness to boost motivation and engagement through its incorporation. Moreover, participants in the second phase of the pre-test were presented with challenging tasks tailored to their specific roles. Designers were assigned the task of creating a visually stunning and user-friendly landing page for a travel booking website, focusing on captivating imagery, intuitive navigation, and persuasive call-to-action elements. On the other hand, developers were tasked with developing a responsive and feature-rich mobile application for a fitness tracking platform, incorporating functionalities such as activity tracking, goal setting, and social sharing. Additionally, testers were assigned the task of

**Table 3 Software professionals' inclination towards gamification.**

| No. | Statements | Scale 1 | 2 | 3 | 4 | 5 | Mean | Standard deviation |
|---|---|---|---|---|---|---|---|---|
| 1. | I possess prior knowledge about gamification. | 2 | 5 | 11 | 23 | 9 | 3.64 | 1.02 |
| 2. | I hold a positive view regarding the integration of gamification into workplace practices. | | | 8 | 33 | 9 | 4.02 | 0.58 |
| 3. | I believe gamification has the potential to enhance my effectiveness in professional tasks. | | 3 | 9 | 26 | 12 | 3.94 | 0.81 |
| 4. | Gamification can create a more dynamic and effective work environment. | | 4 | 12 | 21 | 13 | 3.86 | 0.89 |
| 5. | I am open to adopting gamification techniques at work if they are introduced by management. | | | 16 | 19 | 15 | 3.98 | 0.79 |
| 6. | I would appreciate it if gamification methods were incorporated into our workplace processes. | 1 | 4 | 12 | 15 | 18 | 3.90 | 1.04 |
| 7. | I have a preference for both individual and collective competitive gamification activities. | | 9 | 8 | 24 | 9 | 3.66 | 0.97 |
| 8. | I think utilizing gamification would add an element of enjoyment to work-related activities. | | 3 | 9 | 22 | 16 | 4.02 | 0.86 |
| 9. | Gamification is likely to boost my engagement in professional learning. | | 7 | 6 | 26 | 11 | 3.82 | 0.93 |
| 10. | Some colleagues and superiors have already integrated gamification into their work approaches. | 19 | 26 | 5 | | | 2.72 | 0.63 |

**Note:**
1- Strongly Disagree. 2- Disagree. 3- Neutral. 4- Agree. 5- Strongly Agree.

rigorously evaluating the mobile application designed by the developers, ensuring functionality, usability, and performance standards were met, and identifying any bugs or areas for improvement. Each task demanded a combination of creativity, technical expertise, and attention to detail. Evaluations were conducted by industry based on a maximum score of 60, assessing criteria such as design aesthetics, coding standards, functionality, and user experience. Success in the task was determined by exceeding predefined benchmarks, showcasing proficiency and innovation in their respective domains. Figure 8 presents the results of the software professionals, and the outcomes suggest that the software professionals share an equivalent level of expertise.

## Treatment

This section discusses the treatment process applied to the groups participating in the experiment. Each group pursued the designated treatment individually, based on their respective methodologies. Furthermore, these groups operated within a provided work environment for a duration of 1 year. The CEOs of the IT organizations closely monitored the entire work environment. Additionally, the researcher offered guidance to professionals, addressing their concerns throughout this process. Group-wise descriptions of the treatments are provided below: Control group: This group adhered to conventional working methods and received no experimental interventions. They followed the standard workplace practices without incorporating any technological changes. Their daily tasks and monthly projects aligned with traditional practices. Intervention group: This group engaged with a gamified website developed by the researcher as part of the experimental treatment. Despite their proficiency in using websites, all software professionals receive detailed demonstrations on navigating the gamified website for their daily tasks. Their daily tasks were conducted through this platform. Notably, the intervention group interacted with the gamified website exclusively for their daily tasks.

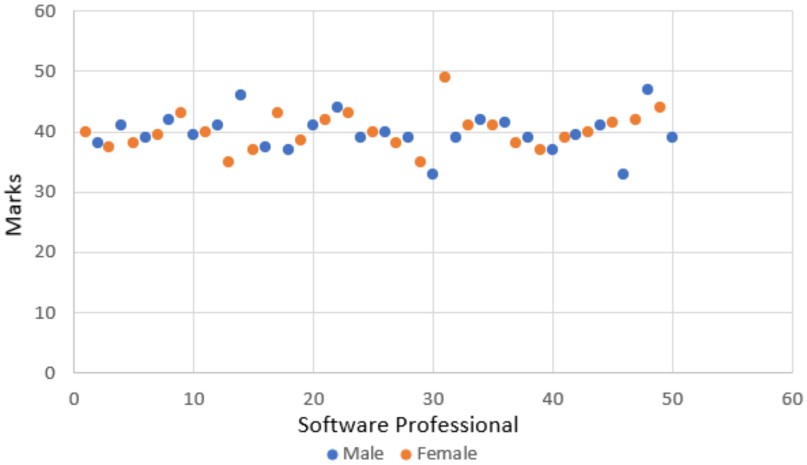

**Figure 8 Pre-test task scores of software professionals.**

## Post-test

Following the treatment phase, both groups commenced their work within their respective assigned environments. The control group adhered to traditional work practices for task completion, while the intervention group embraced the gamified website as a tool for their day-to-day work activities. Consequently, all accomplished tasks were submitted for review to team leaders or CEOs. This post-test phase enabled the evaluation of how the different methodologies adopted by each group, traditional *vs* gamified, influenced their task execution and subsequent assessment by higher authorities.

## RESULTS AND DISCUSSION

This section presents and discusses the results of both groups involved in the experiment through two distinct methods: 1) tabular representation indicating average scores, and 2) the application of statistical analysis.

## Results

The results section presents a clear and compelling illustration of the superior performance of the intervention group over the control group in achieving daily targets. Statistical and tabular analysis validates the significance of these findings, enhancing the credibility and rigor of the research outcomes.

### *Tabular analysis*

Table 4 presents the data analysis of the average outcomes for both groups in achieving their daily targets. The findings indicate that the intervention group demonstrates more pronounced progress than the control group. During the 1-year experimental period, both the intervention group and the control group underwent comprehensive evaluations across multiple metrics, including task completion, task score, code quality, timeliness of task completion, new courses learned, and average team task performance. These metrics were meticulously selected to assess the overall motivation, development, and engagement levels within each group.

**Table 4  Post-test results.**

|  | Group | Mean | SD | Skewness | Kurtosis | Std. Error mean |
|---|---|---|---|---|---|---|
| Task completed | Intervention group | 15.80 | 3.014 | 0.298 | −0.933 | 0.60 |
|  | Control group | 12.24 | 1.786 | 0.704 | −0.246 | 0.357 |
| Task score | Intervention group | 143.72 | 29.365 | 0.682 | −0.459 | 5.873 |
|  | Control group | 103.16 | 15.367 | 1.152 | 1.576 | 3.073 |
| Code quality | Intervention group | 71.04 | 15.380 | 0.032 | −0.963 | 3.076 |
|  | Control group | 53.32 | 10.535 | 1.152 | 1.075 | 2.107 |
| Timeliness of task | Intervention group | 13.84 | 3.436 | 0.375 | −1.250 | 0.687 |
|  | Control group | 8.80 | 1.354 | 0.066 | −1.101 | 0.271 |
| New course | Intervention group | 3.84 | 1.724 | −0.527 | −1.024 | 0.820 |
|  | Control group | 1.28 | 0.980 | 0.245 | −0.842 | 1.578 |
| Average score in team tasks | Intervention group | 64.14 | 4.099 | −0.393 | −0.620 | 0.345 |
|  | Control group | 52.46 | 7.890 | −1.173 | 0.784 | 0.196 |

Analysis of the data presented in Table 4 revealed significant findings. When considering mean and standard deviation values, it became apparent that the intervention group consistently outperformed the control group across all evaluated metrics. For instance, the intervention group exhibited a mean of 15.80 tasks completed, compared to the control group's mean of 12.24 tasks. Similarly, in terms of task scores, the intervention group achieved an average of 143.72, while the control group averaged 103.16. This trend was consistent across all other metrics examined. These results underscore the effectiveness of employing gamification strategies in enhancing the motivation, development, and engagement levels of software professionals. The observed superiority of the intervention group highlights the substantial impact of integrating gamification methodologies to foster productivity and engagement in professional settings.

### Statistical analysis

Statistical analysis is a fundamental tool in research that enables researchers to derive meaningful insights from data (*Dixon & Massey, 1951*). It involves the application of mathematical and computational techniques to analyze collected data, helping to uncover patterns, relationships, and trends within the dataset. Through statistical analysis, researchers can draw quantified conclusions about processes or phenomena, enhancing understanding of underlying dynamics. Two commonly used techniques in statistical analysis are the t-test and Analysis of Variance (ANOVA) (*Achen, 2021*). The t-test is used to compare means between two groups, while ANOVA extends this comparison to more than two groups, identifying significant differences. In this research, we employ the t-test to specifically evaluate the distinctions between two distinct groups: the control and intervention groups. This approach allows us to gauge the significance of disparities observed between these two groups effectively.

Throughout the experiment duration, software professionals in both groups were tasked with multiple assignments, each evaluated based on task completion status, attained scores,

work quality, and timeliness of submission. Additionally, professionals' receptiveness to learning new technologies and their performance in team-oriented tasks were assessed. A comprehensive dataset was compiled and subjected to analysis using the IBM SPSS Statistics tool. Table 4 of our study presents an extensive examination of this data, comparing the intervention group to the control group across various critical variables. Statistical techniques, including mean, standard deviation (SD), skewness, kurtosis, and standard error means, were employed to uncover different dimensions of the data and provide valuable insights into group performance. The mean offered an aggregate view of average performance levels across different metrics, while standard deviation measured the extent of variation within each group's performance. Skewness and kurtosis illuminated the distribution and shape of the data, respectively, while standard error means gauged the accuracy of population mean estimates. Through this meticulous statistical analysis, we gain a deeper understanding of the experiment outcomes. The results have shown that the intervention group outperformed the control group in various aspects, including task completion, task scores, timeliness, code quality, team tasks, and enrollment in new courses. Moreover, insights from standard deviation and standard error of the mean results provide further clarity on the consistency and reliability of our findings, suggesting a positive impact on performance and skill development compared to the control group.

The research first conducted Levene's test to assess whether the variances of two or more groups are equal or not. It calculates the deviations of individual data points from their respective group means and compares the variability between groups. In Levene's test, the variable F represents the test statistic, which measures the ratio of the variance between groups to the variance within groups. A larger F value indicates greater variability between groups relative to within groups. The variable Sig. (short for significance) represents the *p*-value associated with the test statistic. It indicates the probability of observing the test results, assuming that the null hypothesis (*i.e.*, equal variances between groups) is true. If the *p*-value (Sig.) is less than the chosen significance level (typically 0.05), it suggests that there is a significant difference in variances between groups. Furthermore, to rigorously assess and validate the observed differences between the intervention group and the control group, a t-test was conducted. The t-test serves as a statistical tool to monitor the performance of both groups and evaluate the significance of the disparities in their results based on test statistics (t) and degree of freedom (df). The outcomes of this t-test analysis were utilized to validate and verify specific hypotheses:

**H1**: Gamification positively influences task completion when comparing the intervention group to the control group

**H2**: Gamification positively influences task performance when comparing the intervention group to the control group

**H3**: Gamification positively influences code quality when comparing the intervention group to the control group

**H4**: Gamification positively influences on-time work completion when comparing the intervention group to the control group

**Table 5 t-test result for the total task completed.**

| | | Levene's test for equality of variances | | t-test for equality of means | | | | | 95% confidence interval of the difference | |
|---|---|---|---|---|---|---|---|---|---|---|
| | | F | Sig. | t | df | Sig. (2-tailed) | Mean diff. | Std. error diff. | Lower | Upper |
| Task completed | Equal variances assumed | 7.022 | 0.011 | 5.081 | 48 | 0.000 | 3.560 | 0.701 | 2.151 | 4.969 |
| | Equal variances not assumed | | | 5.081 | 39.006 | 0.000 | 3.560 | 0.701 | 2.143 | 4.977 |

**H5**: Gamification positively influences new learning when comparing the intervention group to the control group

**H6**: Gamification positively influences task performance during group tasks when comparing the intervention group to the control group

These hypotheses were formulated to test the significance of the differences in various aspects of performance and skill development between the two groups, providing a comprehensive evaluation of the effectiveness of the gamified working methods. In order to rigorously evaluate these hypotheses and ascertain the extent of the differences between the control and intervention groups, we conducted a series of independent sample t-tests. The results of these t-tests are presented in Tables 5 through 10. These tables illustrate the statistical findings pertaining to total tasks completed, scores gained, code quality, timeliness of task completion, new courses learned, and average scores in team tasks. These tests have been instrumental in providing empirical evidence and a quantitative understanding of the effectiveness of the intervention group, shedding light on the specific domains in which gamification has influenced performance and skill development in comparison to traditional working methods. The results of the independent samples t-tests affirm the validity of our defined hypotheses regarding the impact of gamification on various aspects of performance and skill development when comparing the intervention group to the control group.

## Hypothesis 1: impact on task completion

The analysis begins by formulating a null hypothesis against Hypothesis 1 (H1): The null hypothesis is formulated as: H0a: Gamification does not positively influence task completion when comparing the intervention group to the control group while the H1 is defined as H1: Gamification positively influences task completion when comparing the intervention group to the control group The first hypothesis was tested based on the value of the variable task completed. This variable considers the values of the total number of tasks completed during the experiment duration. Upon conducting a t-test, it was observed that the $p$-value (Sig.) is less than 0.05. Consequently, the null hypothesis is rejected, and H1 is deemed valid. Therefore for task completion, participants in the intervention group

**Table 6  t-test result for the task scores.**

| | | Levene's test for equality of variances | | t-test for equality of means | | | | | | |
|---|---|---|---|---|---|---|---|---|---|---|
| | | F | Sig. | t | df | Sig. (2-tailed) | Mean Diff. | Std. Error Diff. | 95% confidence interval of the difference | |
| | | | | | | | | | Lower | Upper |
| Task score | Equal variances assumed | 11.284 | 0.002 | 6.119 | 48 | 0.000 | 40.560 | 6.629 | 27.232 | 53.888 |
| | Equal variances not assumed | | | 6.119 | 36.228 | 0.000 | 40.560 | 6.629 | 27.120 | 54.000 |

excelled significantly, surpassing the control group by a substantial margin with a mean difference of 3.560 ($t(48) = 5.081$, $p < 0.001$, 95% CI [2.143, 4.977]) as shown in Table 5.

### Hypothesis 2: impact on task performance

The analysis of the second hypothesis initiates with the formulation of its corresponding null hypothesis, denoted as H0b: Gamification does not positively influence task performance when comparing the intervention group to the control group against the corresponding hypothesis 2 defined as H2: Gamification positively influences task performance when comparing the intervention group to the control group The t-test results shows that since $p$-value is less than 0.005, therefore the null hypothesis stands rejected while the hypothesis 2 is valid where the intervention group achieved notably higher scores, outperforming the control group by 40.560 points ($t(48) = 6.119$, $p < 0.001$, 95% CI [27.232, 53.888]) as shown in Table 6.

### Hypothesis 3: impact on code quality

The analysis of the third hypothesis commences by formulating its corresponding null hypothesis, denoted as H0c: Gamification does not positively influence code quality when comparing the intervention group to the control group, in contrast to Hypothesis 3, defined as H3: Gamification positively influences code quality when comparing the intervention group to the control group. The hypothesis was evaluated based on the value of code quality attained by the software professionals based on the quality of the tasks submitted. Upon conducting the t-test, the results reveal that the $p$-value is less than 0.005. Consequently, the null hypothesis is rejected, affirming the validity of Hypothesis 3. This suggests that gamification indeed has a positive impact on code quality when comparing the intervention group to the control group where the intervention group displayed significant improvement, evident in the mean difference of 17.720 ($t(48) = 4.753$, $p < 0.001$, 95% CI [10.224, 25.216]) as shown in Table 7.

### Hypothesis 4: impact on on-time task completion

The examination of the fourth hypothesis begins with formulating its corresponding null hypothesis, designated as H0d: Gamification does not positively influence on-time work

**Table 7 t-test result for the code quality.**

| | | Levene's test for equality of variances | | t-test for equality of means | | | | | | |
|---|---|---|---|---|---|---|---|---|---|---|
| | | F | Sig. | t | df | Sig. (2-tailed) | Mean Diff. | Std. Error Diff. | 95% confidence interval of the difference | |
| | | | | | | | | | Lower | Upper |
| Code quality | Equal variances assumed | 4.891 | 0.032 | 4.753 | 48 | 0.000 | 17.720 | 3.728 | 10.224 | 25.216 |
| | Equal variances not assumed | | | 4.753 | 42.457 | 0.000 | 17.720 | 3.728 | 10.198 | 25.242 |

**Table 8 t-test result for the timeliness of task completed**

| | | Levene's test for equality of variances | | t-test for equality of means | | | | | | |
|---|---|---|---|---|---|---|---|---|---|---|
| | | F | Sig. | t | df | Sig. (2-tailed) | Mean diff. | Std. error diff. | 95% confidence interval of the difference | |
| | | | | | | | | | Lower | Upper |
| Timeliness of task completed | Equal variances assumed | 34.798 | 0.000 | 6.823 | 48 | 0.000 | 5.040 | 0.739 | 3.555 | 6.525 |
| | Equal variances not assumed | | | 6.823 | 31.278 | 0.000 | 5.040 | 0.739 | 3.534 | 6.546 |

completion when comparing the intervention group to the control group. This null hypothesis is juxtaposed against Hypothesis 4, articulated as H4: Gamification positively influences on-time work completion when comparing the intervention group to the control group. This hypothesis was substantiated using data collected on the timeliness of task completion by software professionals. Upon analysis employing the t-test, the results unveil a *p*-value below 0.005. Consequently, the null hypothesis is rejected, affirming the validity of Hypothesis 4. This indicates that gamification acts as a catalyst for improving on-time work completion when contrasting the intervention group with the control group. Additionally, the intervention group exhibited a notable advantage in timeliness of task completion, with a significant difference of 5.040 s ($t(48) = 6.823$, $p < 0.001$, 95% CI [3.555, 6.525]) as shown in Table 8.

## Hypothesis 5: impact on new learning

The fifth hypothesis investigation begins with formulating its corresponding null hypothesis, denoted as H0e: Gamification does not positively influence new learning when comparing the intervention group to the control group. Contrary to this, Hypothesis 5, labeled as H5, posits that: Gamification positively influences new learning when comparing the intervention group to the control group. To evaluate this hypothesis, data

**Table 9  t-test result for the new course learned.**

| | | Levene's test for equality of variances | | t-test for equality of means | | | | | | |
|---|---|---|---|---|---|---|---|---|---|---|
| | | F | Sig. | t | df | Sig. (2-tailed) | Mean Diff. | Std. Error Diff. | 95% confidence interval of the difference | |
| | | | | | | | | | Lower | Upper |
| New course learned | Equal variances assumed | 7.893 | 0.007 | 6.454 | 48 | 0.000 | 2.560 | 0.397 | 1.762 | 3.358 |
| | Equal variances not assumed | | | 6.454 | 38.035 | 0.000 | 2.560 | 0.397 | 1.757 | 3.363 |

on software professionals' engagement in new learning activities as well as the total number of new languages or technologies learned were collected and analyzed. The t-test results reveal a *p*-value below 0.005. Consequently, the null hypothesis is rejected, supporting the validity of Hypothesis 5. This indicates that gamification indeed plays a role in fostering new learning among participants in the intervention group compared to those in the control group with a mean difference of 2.560 (t(48) = 6.454, $p < 0.001$, 95% CI [1.762, 3.358] as shown in Table 9).

## Hypothesis 6: impact on team task performance

The examination of the sixth hypothesis begins by formulating its corresponding null hypothesis, designated as H0f: Gamification does not positively influence task performance during group tasks when comparing the intervention group to the control group. This null hypothesis is juxtaposed against Hypothesis 6, articulated as H6: Gamification positively influences task performance during group tasks when comparing the intervention group to the control group. To scrutinize this hypothesis, data regarding task performance during group tasks were collected and analyzed. Upon conducting the t-test, the findings reveal a *p*-value below 0.005. Consequently, the null hypothesis is refuted, lending support to the validity of Hypothesis 6. This suggests that gamification indeed enhances task performance during group tasks among participants in the intervention group compared to those in the control group which is reflected with a substantial difference of 11.6800 (t(48) = 6.568, $p < 0.001$, 95% CI [8.1047, 15.2553]) as shown in Table 10.

The significant differences revealed through the independent sample t-tests provide strong evidence that gamification has a profound impact on the development, engagement, and motivation of software teams. In each evaluated aspect, such as task completion, task score, timeliness of task completion, code quality, average scores in team tasks, and new course uptake, the intervention group consistently outperformed the control group. These results, with *p*-values consistently below 0.001, demonstrate the statistical significance of the differences between the two groups, thereby substantiating our initial hypotheses. This empirical validation underscores the potency of gamification in elevating performance and skill development in software teams, reaffirming its effectiveness and providing valuable insights for further implementation and innovation in this dynamic field.

**Table 10  t-test result for the average score in team tasks.**

| | | Levene's test for equality of variances | | t-test for equality of means | | | | | | |
|---|---|---|---|---|---|---|---|---|---|---|
| | | F | Sig. | t | df | Sig. (2-tailed) | Mean Diff. | Std. Error Diff. | 95% confidence interval of the difference | |
| | | | | | | | | | Lower | Upper |
| Average score in team tasks | Equal variances assumed | 5.225 | 0.027 | 6.568 | 48 | 0.000 | 11.6800 | 1.7782 | 8.1047 | 15.2553 |
| | Equal variances not assumed | | | 6.568 | 36.077 | 0.000 | 11.6800 | 1.7782 | 8.0739 | 15.2861 |

## Discussions

The study is completed in two major parts to address the research questions outlined earlier. Initially, it delves into identifying the essential elements for developing a gamified system tailored explicitly to software teams operating within the IT organization. This primary investigation revolves around pinpointing key components necessary for crafting a culturally tailored, gamified framework aimed at enhancing motivation within software teams.

The TechMark framework, meticulously established through comprehensive analysis and synthesis of existing literature and expert insights, sheds light on critical constituents such as game elements, dynamics, and feedback mechanisms. However, its significance extends beyond the mere identification of these elements. The framework emphasizes meticulously identifying the target audience, which in this case comprises software professionals in IT organizations. By understanding their roles, responsibilities, and challenges, their specific problems can be anticipated and later expected solutions can be proposed. Furthermore, the framework delves into understanding the user motivation inherent in this demographic. It explores the motivational drivers among software professionals, whether they stem from a desire for recognition or reward, achievement, or intrinsic interest in their work. Armed with this understanding, the specific game elements, dynamics, and feedback techniques are tailored to enhance motivation within this audience. This framework offers a blueprint for IT organizations to create tailored motivation-enhancing systems for their software teams.

Additionally, the study's second research question delves into the critical matter of evaluation and measurement. It seeks to establish a robust methodology for assessing the effectiveness of the proposed gamified framework in motivating software teams to actively engage in their daily tasks and ultimately achieve higher performance levels. This evaluation comprehensively examined two groups—one adhering to traditional work practices (control group) and the other utilizing a gamified platform for their daily tasks (intervention group). The evaluation encompassed crucial parameters such as average tasks completed, average scores, average new learning attained, code quality, timeliness of

task submission, and average team task scores. Each group was meticulously assessed based on these performance metrics to unveil their strengths. After scrutinizing the tabular and statistical analyses, a clear trend emerged: the intervention group exhibited significantly superior performance to the control group. The findings illustrated that the intervention group consistently outperformed the control group across all evaluated dimensions. Notably, the intervention group showcased higher averages in tasks completed, scores achieved, new learning accomplished, code quality, task submission timeliness, and team task scores. The impressive performance of the intervention group sheds light on motivation, development, and engagement at work. Their outstanding results suggest that motivation, development, and engagement are closely linked. The success of the gamified platform is clear from the intervention group's significantly higher scores in learning, code quality, and meeting deadlines which shows that the gamified approach fosters an environment where individuals are not only motivated to excel but also exhibit a strong desire to learn and enhance their skills. By integrating game elements into tasks and activities, individuals are incentivized to engage actively, leading to increased enthusiasm for personal growth and development. This heightened willingness to learn and improve contributes to a more dynamic and productive workplace culture, where continuous learning and skill enhancement are valued and encouraged. Furthermore, the impressive teamwork scores of intervention group serve as strong evidence of the platform's effectiveness in enhancing employee engagement. The collaborative success of software professionals in team tasks suggests a heightened level of involvement and connection, indicating a deeper engagement with tasks and colleagues. This ultimately results in a workforce that is not only more motivated but also more developed and engaged. To summarize, these impressive results highlight the connection between motivation, development, and engagement at work. They stress the significance of incorporating gamification to enhance these factors for software professionals in IT organizations, which is critical for organizational advancement.

## CONCLUSION

The significance of internal marketing for employee development and job satisfaction is undeniable, with contented employees playing a pivotal role in enhancing customer satisfaction and overall growth. In the IT field, the fast-paced technology changes make it crucial to motivate and develop employees to meet these challenges continuously. Numerous studies have demonstrated that computer games improve motivation and enhance employee development and engagement, highlighting gamification as a valuable approach for increasing employee motivation through monitoring and rewarding systems. Therefore, in this work, an innovative gamification framework is established, which can be used to develop gamified applications for the software teams of IT organizations. The framework is intended to enhance the overall motivation and engagement of each employee in software teams, contributing to the organization's progress, as employee motivation is directly linked to the organization's growth. Moreover, the framework is practically applied to design and implement a prototype application, with its efficacy validated through a quasi-experimental research design. Later the data collected is

analyzed using an independent t-test. The outstanding results of intervention group, observed against the metrics such as total tasks completed, task scores, new courses learned, code quality, timeliness of completed tasks, and team task scores, distinctly demonstrate the positive impact of gamification within IT organizations. Furthermore, these metrics serve as strong indicators of gamification's positive impact on the development, engagement, and motivation of software teams, affirming its substantial role in improving these critical aspects of team performance and individual growth. Additionally, this research opens new gateways to gamification in internal marketing for researchers, paving the way for further exploration and innovation in this dynamic field.

### Funding
The authors received no funding for this work.

### Competing Interests
The authors declare that they have no competing interests.

### Author Contributions
- Iqra Obaid conceived and designed the experiments, performed the experiments, analyzed the data, performed the computation work, prepared figures and/or tables, authored or reviewed drafts of the article, and approved the final draft.
- Muhammad Shoaib Farooq conceived and designed the experiments, analyzed the data, authored or reviewed drafts of the article, and approved the final draft.

### Ethics
The following information was supplied relating to ethical approvals (*i.e.*, approving body and any reference numbers):

Department of Computer Science, University of Management and Technology, Lahore (UMT-Reg/2022/32-1(1)).

### Data Availability
The raw data and questionnaire are available in the Supplemental Files.

The code is available at GitHub and Zenodo:

- https://github.com/IqraObaid/webversion
- Obaid, I. (2024). Gamified portal. Zenodo. https://doi.org/10.5281/zenodo.11143159.

### Supplemental Information
Supplemental information for this article can be found online at http://dx.doi.org/10.7717/peerj-cs.2285#supplemental-information.

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
