# Peer review of "TechMark: a framework for the development, engagement, and motivation of software teams in IT organizations based on gamification"

_PeerJ Computer Science, doi:10.7717/peerj-cs.2285_

## Round 0.1 · original submission · Major Revisions

Both reviewers point out that the direction and topic of the manuscript is interesting. Yet, the descriptions are too shallow to judge the content. Especially reviewer 1 gives many concrete suggestions where more details on the studies are needed.

**Language Note:** PeerJ staff have identified that the English language needs to be improved. When you prepare your next revision, please either (i) have a colleague who is proficient in English and familiar with the subject matter review your manuscript, or (ii) contact a professional editing service to review your manuscript. PeerJ can provide language editing services - you can contact us at [email protected] for pricing (be sure to provide your manuscript number and title). – PeerJ Staff

·

Basic reporting

Related to the Intro section, I believe that the introduction section adequately guides readers through the different concepts, this is, staff turnover, internal marketing, gamification, proposed framework, ending with the research questions of the study and the structure of the article. However, I suggest that different concepts should appear in different paragraphs. For example, the concept of staff turnover appears in the first paragraph, as does the concept of internal marketing.
Related to the Related Work section, my main concern is whether the literature review is a systematic literature review (SLR) or systematic literature mapping or another thing based on the research questions posed in the intro section. If so, authors could add a summary of the SLR in terms of the process followed, the number of relevant studies (8 or 9) provided in Table 1, etc. In fact, the authors should check the last reference in Table 1, because a different reference (Altomari et al. (2022)) appears in the text (page 4, line 150), and the one of Table 1 is not explained. So, how many relevant papers do they have? Eight or nine?
In the Serious Game Design Model for Software Houses section, on page 5 line 179, the authors refer to Figure 1, but this figure has no relation to what they indicate just before (goal identification, game elements, game mechanics, social interaction, and usability).
In the Flow theory subsection, when explaining Figure 2 (page 6, lines 192-203), authors refer to “Secondly, flow involves the Merging of action and awareness”, but there is no component to refer to with this phrase (“Firstly” is the “intense focus”, and “thirdly”, they refer to “clear goals and feedback”, and “Additionally, a sense of control”).
In Tier 2, the name of the tier and the name indicated in Figure 3 should be the same (Game Design or Game Mechanics and Design).
In Tier 3, the name of the tier and the name indicated in Figure 3 should be the same (User Interface is not indicated in Figure 3).
Related to Figures and Tables. This is probably one of the main issues as most of figures and tables are included before being referenced in the text itself or too far away from its reference, so readers can easily get lost. Some examples are:
• In the case of Figure 1, it appears at the bottom of page 2, and it is referred to in the second line.
• Table 1 is in the middle of a paragraph (top of page 4).
• The reference to Figure 2 (page 6) appears on the page following Figure 2 itself (page 5).
• Figure 3 appears before the subsection it refers to.
• Table 4, Figure 9 and Table 5 are showing the same information. I would suggest deleting Figure 9 and merging Tables 4 and 5.
Related to the English language as well as blank spaces. I suggest being reviewed. Some examples are:
• On page 7, line 223 “said problem”.
• On page 2, line 49, engagementRafiq and Ahmed
• On page 2, line 53, technologiesMirghaderi
• On page 9, line 277, (Blank space) Design engaging challenges

Experimental design

Additional information related to the companies of the software professionals should be provided, for example, how many companies are involved? What are the business sectors? And the size? How many developers, designers and testers are there by company?
In relation to the pre-test phase and taking into account the information provided in Table 3, it can be observed that the number of responses for each question is 40 (for example in question number 5, 11+16+13 = 40), however the authors indicate that there were involved 50 professionals (page 12, line 415).
In Figure 8, there are also 40 scores. One question that arises is whether the task to be completed is assessed out of 100 points. If so, it means that the software professionals could be catalogued as junior as their scores are around 40 points.
Another question that arises is whether the intervention group was trained in the gamified website.
Related to the results and discussion section:
• As indicated above, Figure 9 does not provide more information than indicated in Table 4.
• Revise the data shown in Table 4 and the ones in the section of Tabular Analysis (lines 464, 465 and 468): 15.80 vs 14.98, 12.24 vs 11.5 and 8.80 vs 8.93.
• In the subsection of Statistical Analysis, references (39 and 40) are named in a different format.
• Tables 4 and 5 present the same information except for the columns named as Std. Deviation and Std. Error Mean.
• From line 482 to line 487, the authors are writing what readers can observe in Table 5. Then, a sentence about standard deviations and standard error mean is followed, but the authors do not explain anything about the values of theses variables. In addition, in line 486, data including 3 decimal places are provided.
• Tables 6, 7, 8, 9, 10, and 11 can be placed vertically (instead of horizontally) and acronyms should be explained (e.g., F, Sig., t, df).
• On page 21, line 516, the authors use again the name of intervention (and they are not referring to the group).
• On page 21, from line 522 to 531, the authors should separate the different variables that explain (e.g., task completion, task scores, timeliness of task completion, code quality, and average score) into different paragraphs.
• On page 21, lines 543 to 560, the same as in the previous bullet, the authors should separate the discussion of each research question into different paragraphs.
Very important comment: In the subsection called Discussions, authors refer to “the research questions”, however, the authors refer to them on page 3 line 89 and do not refer to them again in the text until this subsection (page 21). I consider that authors should make reference to them in the methodology section, and in the results and discussion section so that readers can follow the research carried out.

Validity of the findings

I consider that the application of a questionnaire in the pre-test phase together with the tabular analysis and statistical analysis are sufficient to give validity to the results presented.

Additional comments

The title of the paper suggest that it focuses on software teams, so I suggest including “software teams” as an additional keyword.
On page 2 line 45, I understand that the components of internal marketing are taken from the reference Huang (2020), right?.
On page 2, line 66, the sentence “Therefore, improving the organization’s turnover” should be linked to the previous one or a predicate should be added.
On page 6, line 191, the two points, : , are repeated (delete one of them).
On pages 7, 9 and 10, a point should be added after each activity/task (for example, in Tier 1, Identify Objectives”.” Objectives …).
Figure 4 appears on page 8, but it is referred to on page 10.
Table 2 appears at the end of the Proposed Model subsection (page 10), and it is referred in the Validation of the Model subsection (page 11). Please, try not to use synonyms, (raters, on page 11, line 357 and 366; and Evaluators in Table 2)
On page 14 line 421, In the pre-test subsection, delete the point.
On page 14, line 443, the authors refer to the “Intervention group”, however on page 12 line 416, it is called “The experimental group”.
On page 15, Table 3 should be revised. What is “Sr.” and SD in the first row of the table? SD stands for Strongly Disagree? Or perhaps it could be Standard Deviation.
On page 17 line 510, in “providing a comprehensive evaluation of the effectiveness of the intervention in comparison to traditional working methods.”, the authors are referring to both working methods, however the new one is called intervention, like the name of the group, so readers may be confused. Please, revise the sentence.
On page 21, line 560, “560 group showcased higher averages in tasks completed;,scores achieved, new..” delete the “;”
On page 22, lines 587 to 589, review the use of capital letters.

Reviewer 2 ·

Basic reporting

The paper is not consistent; it seems to me that it tries to address different aspects without going deep into any one of them.
(1) They propose a framework to design serious games for IT, but it is not clear how the framework is specific of IT and the evaluation of the framework is very weak.
(2) They do an empirical study on the use of a serious game in IT organizations. This is more interesting, but a more detailed description is required.

See more detailed comments in the attached commented article.

Experimental design

The design of the empirical study for part (2) should be better done, or at least better explained. For part (1) there is a single paragraph in the model evaluation.

See more comments in the attached doc.

Validity of the findings

Due to the lack of detail I am not convinced in the validity of results for (1) and (2), though I think that (2) is more interesting and I suggest that the authors rewrite the paper focusing on (2) only.

See further comments in the attached file.

Additional comments

I suggest a major revision where part (1) is removed and part (2) is deeply improved.

Annotated reviews are not available for download in order to protect the identity of reviewers who chose to remain anonymous.

---

## Round 0.2 · Minor Revisions

Unfortunately, only one of the two previous reviewers was available to re-review your submission. I carefully looked through their review, however, and saw strong improvements in response to their comments. I therefore suggest that you address the last remaining points by reviewer 1 in a minor revision.

·

Basic reporting

The authors have adequately responded to all my concerns, although there are still small aspects to improve.
Regarding Figures, I think there are still some figures whose location in the text should be changed. For example, Figure 2 should be placed after its reference in the text, that is, after the first paragraph of the flow theory section. The same goes for Figure 3, its location in the manuscript should be before the subsection “Tier 1: Goal Identification”. And The reference to Figure 4 appears in the text after the figure itself. Ideally, you should place the reference to the figure in the text before the figure appears in the text.
In the related work section, lines 168 and 169 should be eliminated when they specify what is related to their research, because in the rest of the relevant studies nothing has been said about how the research of the authors of this article is focused on.
Changes relative to the explanation in Figure 2 remain unclear. An order is shown in Figure 2, that is, starting with “intense focus”, and ending with “intrinsic motivation”, however this is not the order of the explanation given by the authors on page 7, lines 224 to 236. It should be revised by authors to clarify it.
When the authors are explaining each of Tiers, they must explain in the text what it is about, i.e., for Tier 1, “Identify the problem and define the expected solution”, “identify the specific goals”, etc., are they activities, steps or what?

Experimental design

The authors have adequately responded to all my concerns, although there are still small aspects to improve.
The authors indicate that the IT professionals are developers, designers and testers, however in the results there is no reference to tester type (page 17, lines 498 – 502). Regarding the gender, male or female, there is no result in this regard. In this case, Figure 8 says nothing about the gender or the type of IT professional.
In the pre-test, the authors refer to the second phase as “the task”, so I consider they should say first phase for the 10-question questionnaire (page 17, line 490).

Validity of the findings

No comments.

Additional comments

Some minor errors could be:
• Review the use of the comparative in the last sentence of the abstract.
• On page 4, line 135, missing an “a” in “with gamified one” (should be “with a gamified one”).
• In Table 1, put a lowercase in “The Lack of employee satisfaction ….” Problem Discussed column and Silic et al. (2020) row.
• The term “software houses” has been changed in the new version by IT organizations, however, the term “software houses” still exists a few times.
• The subsection “Proposed Model” should be renamed as Proposed Framework (since this is how the authors refer to their proposal in the manuscript).
• Again, Tier 3 is called “User interface and experience design” in Figure 3, and the authors put only “experience design” on page 8, lines 252-253. The same for Tier 4, see Figure 3 and page 10, line 374.
• On page 8, line 263, remove SAID in “said problem”. Said is related to the verb say, said, said. THIS IS NOT AN ADJECTIVE (this is the second time I refer to this).
• On page 9, line 284, put lowercase in “Identify Objectives” as in the rest of the sentences.
• In Table 2, remove the “:” in “Inter-Rater Reliability Mean:”.
• On page 18, line 526, “leads” should be “leaders”.
• On page 26, line 755, add a “.” In “…an independent t-test The outstanding …”.

---

## Round 0.3 · accepted · Accept

All remaining minor issues have been addressed in this revision. It is ready for publication!